# BERTwich: Extending BERT's Capabilities
# to Model Dialectal and Noisy Text

**Aarohi Srivastava** and **David Chiang**
Computer Science and Engineering
University of Notre Dame
Notre Dame, IN, USA
{asrivas2, dchiang} @nd.edu

## Abstract

Real-world NLP applications often deal with nonstandard text (e.g., dialectal, informal, or misspelled text). However, language models like BERT deteriorate in the face of dialect variation or noise. How do we push BERT's modeling capabilities to encompass nonstandard text? Fine-tuning helps, but it is designed for specializing a model to a task and does not seem to bring about the deeper, more pervasive changes needed to adapt a model to nonstandard language. In this paper, we introduce the novel idea of sandwiching BERT's encoder stack between additional encoder layers trained to perform masked language modeling on noisy text. We find that our approach, paired with recent work on including character-level noise in fine-tuning data, can promote zero-shot transfer to dialectal text, as well as reduce the distance in the embedding space between words and their noisy counterparts.

## 1 Introduction

Pre-trained language models such as BERT (Devlin et al., 2019) contain large amounts of knowledge within their numerous parameters but require relatively low computational power to fine-tune on a downstream task, making them the state of the art for a wide range of popular natural language processing (NLP) tasks. However, evaluation settings and metrics are not always representative of real world use cases. One recurring problem is the severe deficit in performance when BERT-based models are used with *noisy text*.

We define noisy text as any text that deviates from the pre-training data of the model by way of orthographic and/or grammatical variation. The possible sources of this variation are diverse and include dialect variation, user errors (typos, misspellings, grammatical errors), and transcription errors (propagated from errors in optical character recognition or automatic speech recognition systems). We aim to understand how BERT's modeling effectiveness

deteriorates in the face of noisy text and develop novel methods to improve BERT's modeling of noisy text.

Studies have shown that humans can easily read text with high amounts of noise (Landauer et al., 1997). It is curious, then, that BERT's accuracy on downstream tasks drops dramatically when noise is added to test data (Kumar et al., 2020; Yin et al., 2020; Aspillaga et al., 2020). This seeming contradiction comes down to the way in which BERT's textual input is expressed (subword tokenization) and the resultant units to which BERT assigns contextual embeddings (subwords). For instance, take the sentence and corresponding tokens after applying the BERT-base uncased tokenizer:

I am a student $\rightsquigarrow$ I, am, a, student

If there were a single typo in the topical cue *student*, the corresponding tokens would be as follows:

I am a studebt $\rightsquigarrow$ I, am, a, stud, eb, t

The subword tokenizer's rigidity manifests in two ways: *over-segmentation* and *subword replacement* (Kumar et al., 2020; Soper et al., 2021). In the presence of noise, words get *over-segmented* into shorter-length tokens (*stud*, *eb* and *t* above), which may have less meaningful embeddings or even change the meaning of the sentence (*stud* above). These two roadblocks hinder the model from arriving at the correct meaning of a noisy sentence, particularly when, as above, the noise falls on a word with important content. In fact, we find that the distance between the vector encodings of words and their noisy counterparts is extremely high in BERT's embedding space, and we aim to minimize this distance (Table 5).

In this work, we aim to make BERT-based models more robust to the issues of over-segmentation and subword replacement by exposing the models to text with induced character-level perturbations at various stages of training. Previous efforts (see Section 2) have demonstrated the effectiveness of

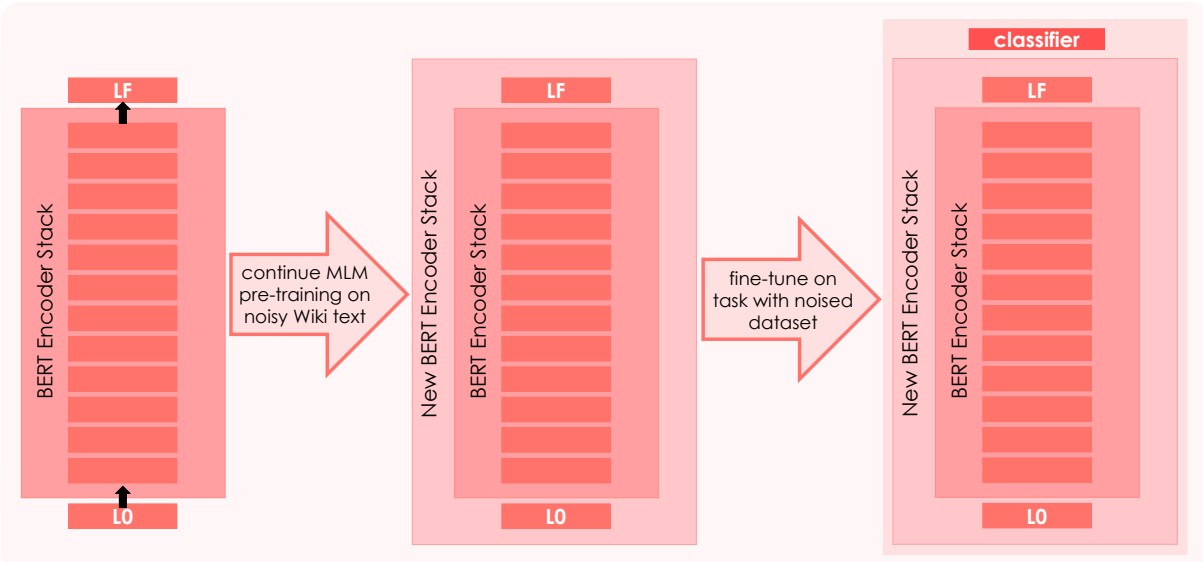

Figure 1: A sketch of our method, BERTwich. We sandwich BERT's encoder stack between prepended and appended randomly-initialized encoder layers, L0 and LF. We continue the masked language modeling pre-training of this expanded encoder stack on synthetically-noised Wikipedia text. The BERTwich model can be fine-tuned on a downstream task with noised fine-tuning data to evaluate the model in settings with dialectal variation or noise in text.

introducing noise either during training or fine-tuning. But here we face a conundrum. On the one hand, training on noisy data from scratch is not practical with larger language models. On the other hand, fine-tuning is designed for more task-specific learning rather than pervasive changes to BERT's language modeling capabilities.

In this paper, we propose that the way out of this conundrum is to include additional Transformer encoder layers (prepended and/or appended to BERT's encoder stack) before continuing pre-training the full model for a few more epochs on synthetically-perturbed Wikipedia text (Figure 1). We call this method *BERTwich*.

We evaluate our method on nonstandard text (from language varieties or dialects unseen during training) as well as synthetically-noised text (i.e., typo simulation). We find that sandwiching BERT between added layers optimized for noisy text modeling improves BERT's performance on text from unseen dialects and reduces the distance in the embedding space between words and their noisy counterparts, as well as the distance between sentence representations of standard and nonstandard text. Our results indicate that without the additional layers, the model would have to re-learn representations of *stud*, *eb*, and *t* that are equivalent to the representation of *student*, while our method makes it easier for the model to simply learn to map the

shorter-length tokens to the existing representation of *student*.

## 2 Related Work

The detrimental effect of noisy test data on BERT's performance is well-documented (Kumar et al., 2020; Yin et al., 2020; Aspillaga et al., 2020; Wu et al., 2022). Synthetically-induced errors (typos, misspellings, grammatical errors, and lexical changes) significantly decrease performance across a medley of tasks and test cases. Moreover, past findings exhibit a large performance gap when evaluating on standard and nonstandard varieties of a language (Aepli and Sennrich, 2022; Srivastava and Chiang, 2023; Held et al., 2023). As discussed above, the source of these issues boils down to the rigid relationship between subword tokenization and resultant subword token embeddings (Kumar et al., 2020; Soper et al., 2021).

Three general approaches have been taken in past work to address this issue of noisy text modeling. Some have tried to tackle it as a text normalization problem during preprocessing (Han et al., 2013; Supranovich and Patsepnia, 2015; Benamar et al., 2021; Demir and Topcu, 2022). Others have attempted to get to the root of the problem by improving the tokenization algorithm or using a character-based model (Hofmann et al., 2021; Lee and Shin, 2021; Tay et al., 2021; Wang et al., 2021).

The third approach is to inject synthetic noise in training, which we find to be the most flexible and generalizable while requiring a low computational load, and our method fits in this final category of solutions. By exposing the model to noise during some stage of training, the model is expected to be better-equipped to deal with noisy text during inference. For instance, Karpukhin et al. (2019) include synthetic noise while training machine translation systems and find performance improvements in evaluation settings with natural noise; however, the approach depends on training a model from scratch rather than adapting an existing model like BERT. Using this approach in fine-tuning, past work has found augmenting the task training data with noisy or adversarial examples and fine-tuning the model with the augmented dataset to provide improvements in performance on noisy text (Vaibhav et al., 2019; Yin et al., 2020).

Applying this approach to dialects, Aepli and Sennrich (2022) inject synthetic character-level noise in continued pre-training and fine-tuning steps for various tasks and find that part-of-speech tagging on dialectal text improves. Followup work by Blaschke et al. (2023) develops methods of predicting the effectiveness of cross-lingual transfer based on the amount of noise injected. Srivastava and Chiang (2023) extend the method to strictly zero-shot settings and apply it to various sequence classification tasks, finding that the approach works well in tasks that can be solved using surface-level cues, but it is not as helpful in more challenging tasks, indicating that more can be done to truly improve the modeling of nonstandard text.

A related but distinct approach involves adding adapter layers tuned to nonstandard text (Pfeiffer et al., 2020; Held et al., 2023). Our method is similar to this work when it comes to adding parameters to the model beyond those used in fine-tuning. However, the methods differ in terms of their functionality; while adapter-based approaches are typically specific to a set of language varieties, our approach provides general improvement to the language modeling capabilities of a BERT-like model with wider applicability to various dialects and to noisy text.

## 3 Method

As demonstrated by previous work, fine-tuning is a powerful method of domain adaptation and cross-lingual transfer. However, while the model may adapt to performing a task on noisy text via fine-tuning, this does not speak to truly extending the model's capabilities and allowing it to better model noisy text.

Starting with a standard pre-trained model such as BERT, our approach has two stages: 1) the BERTwich method: add untrained layers and continue pre-training on noisy data (Sections 3.1 to 3.3), and (2) fine-tune on a specific task (Section 3.4).

### 3.1 Model Variations

We propose a set of BERT variants with modified encoder stacks and additional pre-training in order to better model noisy text. After pre-training, we expand BERT-base by prepending and appending additional layers that are identical to the 12 existing layers, but are randomly initialized. Our experiments include four variations (see Figure 1):

1. BERT + Layer 0 (L0 + BERT): a *blank* encoder layer (with randomly initialized parameters) prepended to the bottom of BERT's encoder stack, resulting in 13 encoder layers.

2. BERT + Top Layer (BERT + LF): a blank encoder layer appended to the top of BERT's encoder stack, resulting in 13 encoder layers.

3. BERT in a Sandwich (L0 + BERT + LF): two blank encoder layers, one prepended and one appended to BERT's encoder stack, resulting in 14 encoder layers.

4. Top-Heavy BERT (BERT + LFx2): two blank encoder layers appended to BERT's encoder stack, resulting in 14 encoder layers.

We follow Devlin et al. (2019)'s original implementation to randomly initialize the new encoder layers.

### 3.2 Continued Pre-Training

Each of the four model variants then undergoes continued pre-training (CPT) on synthetically perturbed text. For comparison, we also perform CPT on BERT-base without making any modifications to the architecture. Our continued pre-training approach is as follows: taking a small random sample of the Wikipedia corpus (approximately 10%), we introduce synthetic character-level noise in the text (see Section 3.3 and 4.1 for details). We train each model on the synthetically-noised Wikipedia text with the masked language modeling objective (with 15% of tokens masked), adopting the same approach

as was used in the original implementation of BERT pre-training (Devlin et al., 2019).

### 3.3 Noising Technique

Our noising technique draws from past work (Karpukhin et al., 2019; Aepli and Sennrich, 2022; Srivastava and Chiang, 2023). We define a word as a substring comprised only of letters (identified using Python's `isalpha` function). We use four noise operations, as done by Karpukhin et al. (2019): *insertion*, *deletion*, *replacement*, and *swapping*. Noise is applied at positions that are part of a word in the input string. We leave non-words (for example, numbers, symbols, and punctuation) unchanged, as we expect modeling capabilities to primarily be affected by linguistic content.

For each letter in a word of the input, noise is applied with probability $p$. We refer to $p$, expressed as a percentage, as the *noise level*. When applying noise, all four noise operations have an equal probability. For the insertion and replacement operations, we randomly select the additional character from the alphabet of the language of the text (e.g., umlaut vowels and eszett are included for German). All random selections are uniform within the set of possibilities. In our continued pre-training, the noise level $p$ is set to 5% per preliminary experimentation. We vary the noise level in fine-tuning at 0, 10, 20, 30, and 40%.

Below, we include a few possible results after applying our noising technique at varying noise levels to an example sentence:

- 0%: colorless green ideas sleep furiously
- 10%: colorless green ideas sloeep furiously
- 20%: colorjless geen ideas sleep fruiouszy
- 30%: ccozorsless greenidesa sleep urruosly
- 40%: colorlses green izeas jsee furkizusly

### 3.4 Fine-Tuning

After doing continued pre-training, we add a linear fine-tuning layer, and we fine-tune the resultant model on one of the tasks described in Section 4.1. Following Srivastava and Chiang (2023), in an effort to not only include noise in the fine-tuning, but also expose the model to multiple variations in the spelling and tokenization of the same words, we use a *joint composition* to set up the fine-tuning data. In the joint composition, the model is fine-tuned

with two copies of the fine-tuning data: one is the original copy, and one is noised (Srivastava and Chiang, 2023). We fine-tune the models five times using one of five distinct noise levels to prepare the noised copy: 0% (baseline), 10%, 20%, 30%, or 40%.

## 4 Experiments

We conduct experiments on sentiment analysis in English with simulated typographical errors, and intent classification in German across multiple dialects.

### 4.1 Data and Tasks

Below, we describe the data used for the continued pre-training, fine-tuning, and evaluation of our models. We work with the English[1] and German[2] uncased BERT-base models.

**English** For continued pre-training, we take a randomly sampled subset of Wikicorpus,[3] which consists of text from English Wikipedia articles. For fine-tuning, we use the Stanford Sentiment Treebank (SST) (Socher et al., 2013), a binary sentiment analysis task that is part of the GLUE benchmark (Wang et al., 2018). The training data consists of 67,300 examples, and the evaluation data consists of 872 examples. In addition to evaluating the models on the SST data itself, we simulate noisy settings in the evaluation data. Specifically, we simulate a one-typo-per-word test scenario using the typo simulation of Naik et al. (2018): for each word in the test data, we select a random character to replace with a letter adjacent to it (left or right) on the standard QWERTY keyboard.

**German** For continued pre-training, we take a randomly-sampled subset of the most recent German Wikipedia dump,[4] which consists of text from German Wikipedia articles. For fine-tuning, we use the German intent classification subset of xSID (van der Goot et al., 2021), a benchmark for cross-lingual slot and intent detection. The xSID dataset was drawn from the English Snips (Coucke et al., 2018) and cross-lingual Facebook (Schuster et al., 2019) datasets and translated to the other languages. The training data (German subset) consists

---

[1] https://huggingface.co/bert-base-uncased
[2] https://huggingface.co/dbmdz/bert-base-german-uncased
[3] https://huggingface.co/datasets/wikicorpus/viewer/raw_en/train
[4] https://dumps.wikimedia.org/dewiki/

| | | Fine-Tuning Noise Level | | | | |
|:---:|:---:|:---:|:---:|:---:|:---:|:---:|
| **Model** | **CPT** | 0% | 10% | 20% | 30% | 40% |
| **BERT** | no | 51.5 ±1.7 | 62.6 ±0.6 | 65.1 ±2.5 | 66.1 ±1.4 | 65.6 ±2.6 |
| **L0 + BERT** | no | 53.2 ±4.7 | 62.8 ±1.7 | 63.8 ±3.1 | 65.2 ±2.5 | 66.1 ±1.0 |
| **BERT + LF** | no | 51.3 ±2.0 | 63.0 ±2.2 | 65.3 ±4.1 | 65.6 ±2.3 | 66.4 ±1.6 |
| **L0 + BERT + LF** | no | 51.4 ±1.3 | 62.4 ±2.1 | 64.5 ±2.0 | 65.8 ±2.0 | 64.2 ±2.5 |
| **BERT + LFx2** | no | 52.4 ±3.2 | 62.2 ±1.6 | 64.0 ±2.2 | 66.9 ±2.1 | 65.9 ±2.0 |
| **BERT** | yes | 56.4 ±3.1 | 70.0 ±0.9 | 72.7 ±1.3 | 73.8 ±1.7 | 73.6 ±1.8 |
| **L0 + BERT** | yes | 57.1 ±2.3 | 70.6 ±1.1 | 73.3 ±2.0 | 74.2 ±2.5 | **74.7** ±1.8 |
| **BERT + LF** | yes | 56.7 ±4.8 | 70.9 ±2.0 | 72.7 ±2.2 | 72.8 ±0.8 | 73.0 ±1.4 |
| **L0 + BERT + LF** | yes | 59.7 ±2.6 | 70.2 ±1.3 | 72.8 ±1.7 | 73.6 ±1.9 | 73.8 ±2.2 |
| **BERT + LFx2** | yes | 58.2 ±1.1 | 70.7 ±1.2 | 72.6 ±2.5 | 73.9 ±0.6 | 73.1 ±2.5 |

Table 1: **English Sentiment Analysis**: Sentiment analysis results with 95% confidence interval measured for five trials, evaluated on one-typo-per-word simulation. The highest score by absolute comparison is in bold. The models following the BERTwich method, particularly L0+BERT, perform better than the baselines and perform best when more noise is used during fine-tuning.

| | | Fine-Tuning Noise Level | | | | |
|:---:|:---:|:---:|:---:|:---:|:---:|:---:|
| **Model** | **CPT** | 0% | 10% | 20% | 30% | 40% |
| **BERT** | no | 79.9 ±3.0 | 90.1 ±0.9 | 93.3 ±0.7 | 93.5 ±1.1 | 95.0 ±0.7 |
| **L0 + BERT** | no | 79.9 ±4.7 | 87.9 ±1.8 | 91.1 ±2.2 | 92.1 ±2.7 | 91.9 ±2.2 |
| **BERT + LF** | no | 80.6 ±5.5 | 87.7 ±1.7 | 91.6 ±2.3 | 94.0 ±2.2 | 94.9 ±1.2 |
| **L0 + BERT + LF** | no | 80.9 ±3.4 | 85.1 ±3.2 | 91.7 ±3.2 | 93.1 ±0.9 | 94.0 ±0.7 |
| **BERT + LFx2** | no | 80.9 ±3.7 | 88.8 ±2.2 | 92.9 ±3.1 | 92.0 ±3.2 | 94.7 ±2.8 |
| **BERT** | yes | 81.6 ±1.3 | 85.6 ±1.7 | 92.3 ±1.4 | 92.9 ±2.1 | 95.1 ±0.9 |
| **L0 + BERT** | yes | 80.7 ±1.5 | 87.4 ±1.1 | 95.2 ±0.9 | 94.7 ±1.6 | 96.2 ±0.6 |
| **BERT + LF** | yes | 79.3 ±0.6 | 87.1 ±2.1 | 93.7 ±1.5 | 96.0 ±0.9 | 96.7 ±1.3 |
| **L0 + BERT + LF** | yes | 83.3 ±2.5 | 91.5 ±1.2 | 97.0 ±0.7 | 96.3 ±0.4 | **97.9** ±0.6 |
| **BERT + LFx2** | yes | 82.7 ±0.9 | 88.5 ±1.4 | 96.5 ±1.7 | 97.7 ±0.9 | 97.2 ±0.3 |

Table 2: **South Tyrolean German Intent Classification**: Results for German BERT with 95% confidence interval measured for five trials, evaluated zero-shot on the South Tyrolean test data. The highest score by absolute comparison is in bold. Though simply fine-tuning with noise greatly boosts performance, the BERTwich models, particularly L0 + BERT + LF, perform better than the baselines.

| | | Fine-Tuning Noise Level | | | | |
|:---:|:---:|:---:|:---:|:---:|:---:|:---:|
| **Model** | **CPT** | 0% | 10% | 20% | 30% | 40% |
| **BERT** | no | 59.7 ±3.9 | 81.7 ±2.3 | 88.7 ±0.7 | **90.1** ±1.4 | 87.9 ±2.8 |
| **L0 + BERT** | no | 62.0 ±3.1 | 80.3 ±3.3 | 87.5 ±1.1 | 88.2 ±2.1 | 87.8 ±2.1 |
| **BERT + LF** | no | 60.2 ±5.3 | 79.7 ±3.9 | 88.7 ±1.3 | 89.9 ±2.1 | 88.5 ±3.0 |
| **L0 + BERT + LF** | no | 65.1 ±6.1 | 78.4 ±4.0 | 89.0 ±1.6 | 89.6 ±1.2 | 88.1 ±2.5 |
| **BERT + LFx2** | no | 60.3 ±5.9 | 81.7 ±3.4 | 89.1 ±2.3 | 89.7 ±2.5 | 88.8 ±2.0 |
| **BERT** | yes | 72.4 ±2.9 | 77.6 ±3.2 | 83.4 ±2.0 | 86.6 ±2.1 | 85.0 ±2.9 |
| **L0 + BERT** | yes | 74.5 ±3.2 | 78.9 ±1.8 | 86.5 ±2.0 | 85.7 ±2.4 | 85.3 ±1.1 |
| **BERT + LF** | yes | 70.0 ±5.0 | 77.2 ±3.1 | 82.9 ±1.4 | 83.9 ±2.7 | 85.9 ±1.3 |
| **L0 + BERT + LF** | yes | 73.7 ±3.4 | 82.1 ±1.6 | 83.4 ±1.3 | 86.1 ±1.4 | 85.2 ±0.8 |
| **BERT + LFx2** | yes | 71.9 ±5.2 | 77.3 ±1.7 | 83.0 ±1.2 | 85.3 ±1.0 | 83.9 ±3.3 |

Table 3: **Swiss German Intent Classification**: Results for German BERT with 95% confidence interval measured for five trials, evaluated zero-shot on the Swiss German test data. The highest score by absolute comparison is in bold. Though fine-tuning with noise without the added layers or CPT yields the highest performance, note that the BERTwich models perform substantially better than fine-tuning alone without noise.

of 10,000 sentences, each labeled with 1 of 18 possible intent classes.

We evaluate our models not only on the standard German test set, but also test sets in two dialects of German:

- South Tyrolean, a Bavarian dialect spoken in the northernmost province of Italy;

- Swiss German, an Alemannic dialect spoken in Switzerland.

We include two examples from xSID (van der Goot et al., 2021) translated in each language. Some instances are closer in appearance across the language varieties, while others vary more substantially.

1. Closer in surface-level appearance:

    - English: Is it cloudy today?
    - German: Ist es heute bewölkt?
    - South Tyrolean: Is heint bewölkt?
    - Swiss German: Isch hüt bewöukt?

2. Farther in surface-level appearance:

    - English: Will it be sunny today?
    - German: Wird es heute sonnig sein?
    - South Tyrolean: Wearts heint sunnig?
    - Swiss German: Schynt hüt d'Sunne?

We do not use any data from the dialects until test time. The test data for each variety consists of 300 sentences.

### 4.2 Baselines and model variations

We compare against several baselines which, unlike our approach, do not alter the BERT architecture.

- BERT models without CPT or added noise in fine-tuning serve as naïve baselines (the straightforward approach of fine-tuning BERT on a task).

- BERT models without CPT but with added noise in fine-tuning follow an approach used to improve performance on noisy and non-standard text in past work (Yin et al., 2020; Srivastava and Chiang, 2023).

- BERT models with CPT and added noise in fine-tuning also follow an approach suggested in past work (Aepli and Sennrich, 2022).

- For completeness, we also compare against BERT with CPT but without added noise in fine-tuning.

Against these baselines, we prepared four models for each language, as described in Section 3.1: L0 + BERT, BERT + LF, L0 + BERT + LF, and BERT + LFx2. We trained all of these models both with CPT (that is, the BERTwich method) and without CPT (that is, the parameters of the new layers remain randomly initialized at the beginning of fine-tuning). We also trained all of these models both with and without added noise in fine-tuning.

### 4.3 Training and testing details

In all of our training runs, we use the AdamW optimizer and a minibatch size of 8. When doing the continued pre-training of a model, we set the learning rate to 1e-4. Continued pre-training takes about 2.9 days on a single NVIDIA A10 Tensor Core GPU. We fine-tune each of the eight models on a sequence classification task (English SST or German xSID) with 5 different noise levels (0, 10, 20, 30, or 40%). When fine-tuning a model, we set the learning rate to 1e-5. We fine-tune each model under each noise level setting five times with a different random initialization each time, and report the average and 95% confidence interval across the five trials.

## 5 Results

The results of our experiments are shown in Tables 1 to 3. Models following the BERTwich approach (added layers + CPT) fall in the final four rows of each table.

### 5.1 English

We fine-tune our English models on the SST sentiment analysis task. Our new models maintain performance with the standard BERT fine-tuning approach on the original SST test data despite our modifications to the model (Appendix A). In Table 1, we show the performance of our models on synthetically-noised SST data (simulating one typo per word). Inclusion of noise in fine-tuning provides a larger jump in performance, and as the fine-tuning noise level increases, so does the performance. For each fine-tuning noise level, our BERTwich models perform the best, and there is a compounding effect of adding the noise-primed layers and injecting higher levels of noise in fine-tuning.

### 5.2 German

We fine-tune our German models on the German subset of the xSID intent classification task. Our

models maintain performance on the test data of the standard variety (German), as desired (Appendix A). We also evaluate them in zero-shot settings on the two dialects of German, which are unseen during any stage of training.

Each model is fine-tuned with a range of noise levels. As shown in Table 2, we find that addition of noise in fine-tuning alone provides boosts in performance on the South Tyrolean test data (a case of zero-shot transfer), and higher noise levels yield the best performance. We attain the highest performance on the South Tyrolean test data with our BERTwich method on the L0 + BERT + LF architecture. This result indicates that our approach can be useful for improving downstream performance on unseen dialects. There is a large jump in performance upon adding noise during fine-tuning, and a steady climb in performance with increasing fine-tuning noise level.

In Table 3, we show the performance of our models on Swiss German test data, another instance of zero-shot cross-lingual transfer. We find that the method of fine-tuning with noise is the most effective in improving intent classification performance on unseen Swiss German data. At the same time, we find that when noise is not present during fine-tuning, the models with CPT perform much better than those without CPT. In fine-tuning tasks where addition of noise may be impractical or unsuitable, continued pre-training with added encoder layers (i.e., the BERTwich method) is a promising approach to greatly boost performance.

## 5.3 Comparison to LoRA

In addition to evaluating several variations of our approach, we also compare BERTwich to LoRA, an alternative to fine-tuning that allows for low-rank adaptation of a Transformer-based language model by adapting the attention weights of each encoder layer and freezing all other parameters when fine-tuning on the task (Hu et al., 2021). We use LoRA in place of vanilla fine-tuning on our two tasks, English sentiment analysis and German intent classification, and apply the approach to two models: BERT and BERT+CPT. We use a learning rate of 5e-4 and set the rank to 8 for our LoRA experiments.

We report the results, averaged over five trials, in Table 4. Just as in the main results, we find that applying more character-level noise to the adaptation/fine-tuning data is beneficial across all

the experiments, and applying LoRA after CPT yields better performance than simply applying LoRA to BERT. At the same time, these results do not outperform our best BERTwich results (Tables 1 to 3).

## 5.4 Embedding Space

To better understand how the embedding space and modeling capabilities change with our approach, we compare the models' representations of standard vs. nonstandard text. We do so in different ways for the English and German models (not fine-tuned). For English (Table 5), we aim to measure the distance in the embedding space between words and their noised counterparts. We do so by applying the one-typo-per-word simulation to the SST test data and comparing the un-noised vs. noised encodings word by word (space-separated sequences). If a word spans more than one subword token, we take the encoding of the word to be the average of the encodings of the tokens that comprise it. We report the averages over all word pairs in the SST test data. For German (Table 6), we compare the sentence representations (i.e., the CLS encoding) for parallel text in standard German vs. one of the two dialects (South Tyrolean and Swiss German). We obtain the parallel text from the xSID intent classification dataset.

We compare the average cosine similarity scores of BERT with those of BERT+CPT to measure the effect of CPT on the embedding space, as well as to the CPT models with added encoder layers to measure the holistic effect of our method. We measure how well the bottom-most layer of each model does this implicit mapping between different versions of the same word, as well as the similarity between the final word embeddings (for English typos) or sentence embeddings (for German dialects) resulting from the top-most layer of the model. For comparison, we also show the average cosine similarity scores for the first and last layer in BERT-base as they are modified by CPT in the other models, even if they no longer remain the bottom- or top-most layers in the new models due to the added layers.

Without any intervention, there is an extremely high distance (i.e., low cosine similarity) in English BERT's embedding space, whether at the bottom or the top layer, between the vector encoding of words and their noised counterparts (Table 5). Though continued pre-training reduces the gap, the addition of encoder layers evidently transforms the role that

| | | Fine-Tuning Noise Level | | | | |
|---|---|---|---|---|---|---|
| **Task** | **CPT** | 0% | 10% | 20% | 30% | 40% |
| **English SA** | no | 51.6 ±1.4 | 62.4 ±1.1 | 64.2 ±1.0 | 65.0 ±2.0 | 64.6 ±0.7 |
| | yes | 58.8 ±1.6 | 67.0 ±1.2 | 69.2 ±0.6 | 69.8 ±1.0 | 69.6 ±2.1 |
| **South Tyrolean IC** | no | 75.2 ±2.4 | 87.2 ±2.7 | 90.4 ±2.9 | 92.6 ±2.9 | 94.0 ±3.2 |
| | yes | 80.8 ±2.0 | 84.0 ±2.3 | 91.2 ±3.7 | 93.8 ±2.0 | 93.6 ±1.4 |
| **Swiss German IC** | no | 59.8 ±5.7 | 76.2 ±5.2 | 85.4 ±3.7 | 86.2 ±2.2 | 86.0 ±1.2 |
| | yes | 73.2 ±1.6 | 78.8 ±3.1 | 83.6 ±1.9 | 84.0 ±2.5 | 84.6 ±4.2 |

Table 4: **LoRA Evaluation**: Results for the English sentiment analysis (SA) and German intent classification (IC) tasks using LoRA rather than vanilla fine-tuning with 95% confidence interval measured for five trials. Our BERTwich approach outperforms LoRA for English SA and South Tyrolean IC, and the two approaches perform the same for Swiss German IC.

| | | **Layer** | | | |
|---|---|---|---|---|---|
| **Model** | **CPT** | L0 | L1 | L12 | LF |
| **BERT** | no | — | 0.13 | 0.24 | — |
| **BERT** | yes | — | 0.27 | 0.54 | — |
| **L0 + BERT** | yes | 0.41 | 0.44 | 0.57 | — |
| **BERT + LF** | yes | — | 0.55 | 0.48 | 0.52 |
| **L0 + BERT + LF** | yes | **0.56** | **0.58** | 0.56 | 0.70 |
| **BERT + LFx2** | yes | — | 0.50 | **0.65** | **0.77** |

Table 5: **Cosine similarity between words with and without typos**: While the cosine similarities between un-noised and noised words in BERT's embedding space are extremely low, they are greatly increased in our BERTwich models, most notably in the lower layers of the model, indicating improved modeling capabilities.

| | | **Dialect** | |
|---|---|---|---|
| **Model** | **CPT** | South Tyrolean | Swiss German |
| **BERT** | no | 0.70 | 0.70 |
| **BERT** | yes | 0.78 | 0.70 |
| **L0 + BERT** | yes | 0.78 | 0.70 |
| **BERT + LF** | yes | 0.79 | 0.73 |
| **L0 + BERT + LF** | yes | **0.85** | **0.80** |
| **BERT + LFx2** | yes | 0.69 | 0.62 |

Table 6: **Cosine similarity between standard and dialectal sentence embeddings (top layer)**: While the cosine similarities between top-layer CLS embeddings of standard vs. dialect parallel sentences in BERT's embedding space are low, they are greatly increased in our BERTwich models (particularly L0 + BERT + LF), indicating improved modeling capabilities.

the bottom layer plays in mapping words exhibiting spelling variation to the standard word, and greatly reduces the gap between representations of words and their noisy counterparts as they would come out from the top of the model. Not only do these results demonstrate the gradual progression in modeling capability of each version of the model, but they also speak to the effectiveness of our contribution of adding uninitialized encoder layers before conducting the continued pre-training with noise.

The results of comparing the distance between the sentence representation (CLS encoding) of a standard German sentence and the parallel dialectal sentence are shown in Table 6, boasting the same finding as for English words that the BERTwich approach brings the internal representation of non-standard text much closer to its standard counterpart than can be found in an undisturbed BERT model. More importantly, this analysis demonstrates the importance of placement and the value of the sandwich model in particular. While BERT + LFx2 contains an equal number of parameters to L0 + BERT + LF, we see that it is no better than the undisturbed BERT at bringing standard and non-standard German text closer together in the internal representation. Rather, the combined efforts of L0 and LF make L0 + BERT + LF superior to the other BERTwich models in its ability to model dialectal text at the sentence level.

## 6 Conclusion

In this paper, we explore the issues surrounding language modeling of nonstandard text and extend BERT's language modeling capabilities to common real-world scenarios involving nonstandard text. Such issues could arise for a variety of reasons, including inference on text exhibiting dialectal variation, user errors such as misspellings or typos,

and propagated errors from optical character recognition and automatic speech recognition systems. We introduce the novel idea of adding randomly-initialized layers to the encoder stack of a pre-trained BERT model before continuing the pre-training of the new model on text with synthetically-induced noise. This way, the newly added layers improve BERT's ability to model nonstandard text. Our approach is called *BERTwich*.

We find that our approach not only improves downstream task performance on nonstandard text, but also expands BERT's modeling capabilities. For instance, we find that the distance in BERT's embedding space between words and their noisy counterparts (e.g., *student* vs. *studebt*) is extremely high, but is dramatically reduced in the embedding space of our BERTwich models. Furthermore, we find that the sentential (CLS) representations within BERT's layers are not equipped to represent dialect text, while those extracted from the upper layers of the best BERTwich model are. Our findings indicate a transformation in the role the lower layer(s) can play in representing noisy text – while BERT would need to re-learn representations of the new token sequence for a noised word, our BERTwich method promotes knowledge transfer from standard to nonstandard text modeling, making it easier for the model to perform mappings to standard variants of words.

## Limitations

In this work, we provide a method of extending BERT's language modeling capabilities beyond what can be achieved by fine-tuning alone, without pre-training a new model from scratch. Our method involves adding layers to BERT's encoder stack and continuing the pre-training of the model for a fraction of the epochs and data than would be needed when pre-training from scratch. Though continued pre-training is far less expensive, it still requires computational power that is available at an institutional level but not always publicly available. As a result, we are interested in identifying ways in which the computational load of our method could be further reduced, possibly using machine learning techniques like model distillation or teacher learning so that the extra layers could be trained in isolation but still be compatible with the full model when added. Along similar lines, we would have liked to conduct multiple trials of the continued pre-training as we did for the fine-tuning to be able

to report statistical measures across the trials.

In an effort to increase the diversity within our experiments, we included results for two languages, English and German, and evaluated our method in two scenarios involving dialectal text and text with typos. Our tasks are both sentence-level classification tasks to allow for a clearer analysis of our methods; however, a more diverse array of languages and tasks would be needed to assess the broader applicability of our method, particularly as it relates to different linguistic features (e.g., types of dialectal variation, morphological structure) and the demands of the downstream task.

## Ethics Statement

Our work involves the modification of existing models using publicly available data, so our research methods themselves do not inherently evoke ethical concerns. The result of our work is an improved ability to model text with features of dialectal variation as well as text with user-generated noise (e.g., misspellings, typos), with the intention to improve the quality of NLP tools for a wider range of users. While this can be a valuable change, one can imagine scenarios in which text that was once poorly handled by NLP systems can be easily comprehended, which may be an undesirable change for some users (e.g., unwelcome monitoring of online communication).

## Acknowledgements

This material is based upon work supported by the US National Science Foundation under Grant No. IIS-2125948.

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

## A  Appendix

| | | Fine-Tuning Noise Level | | | | |
|---|---|---|---|---|---|---|
| **Model** | **CPT** | **0%** | **10%** | **20%** | **30%** | **40%** |
| **BERT** | no | 91.0 ±0.8 | 91.0 ±1.0 | 91.3 ±0.5 | 91.6 ±0.8 | 91.2 ±0.4 |
| **L0 + BERT** | no | 90.3 ±0.5 | 90.7 ±1.0 | 89.2 ±3.9 | 90.8 ±0.9 | 91.4 ±0.3 |
| **BERT + LF** | no | 90.8 ±0.3 | 90.8 ±0.9 | 90.5 ±1.4 | 91.4 ±0.7 | 91.4 ±0.7 |
| **L0 + BERT + LF** | no | 90.5 ±1.3 | 90.7 ±0.5 | 91.0 ±0.8 | 91.3 ±0.8 | 89.2 ±6.1 |
| **BERT + LFx2** | no | 90.6 ±0.6 | 90.5 ±0.5 | 91.4 ±0.7 | 90.8 ±0.5 | 91.3 ±0.8 |
| **BERT** | yes | 88.8 ±1.7 | 88.5 ±0.4 | 88.9 ±0.6 | 89.1 ±1.0 | 88.9 ±0.7 |
| **L0 + BERT** | yes | 88.3 ±1.0 | 88.0 ±0.6 | 88.6 ±0.9 | 89.1 ±0.6 | 88.1 ±0.8 |
| **BERT + LF** | yes | 89.1 ±0.5 | 88.9 ±0.6 | 88.9 ±1.0 | 88.4 ±0.6 | 89.2 ±1.0 |
| **L0 + BERT + LF** | yes | 90.8 ±0.3 | 90.8 ±0.9 | 90.5 ±1.4 | 91.4 ±0.7 | 91.4 ±0.7 |
| **BERT + LFx2** | yes | 89.1 ±0.9 | 88.9 ±1.1 | 89.0 ±0.9 | 89.1 ±1.1 | 89.0 ±1.3 |

Table 7: **English Sentiment Analysis**: Sentiment analysis results with 95% confidence interval measured for five trials, evaluated on the original un-noised SST data. Performance is close to the baseline across all models, as desired.

| | | Fine-Tuning Noise Level | | | | |
|---|---|---|---|---|---|---|
| **Model** | **CPT** | **0%** | **10%** | **20%** | **30%** | **40%** |
| **BERT** | no | 97.5 ±0.8 | 97.8 ±0.8 | 98.3 ±0.6 | 98.1 ±0.7 | 98.2 ±0.5 |
| **L0 + BERT** | no | 97.9 ±0.4 | 97.4 ±0.7 | 98.2 ±0.8 | 98.2 ±0.7 | 98.7 ±0.6 |
| **BERT + LF** | no | 97.8 ±0.9 | 97.9 ±1.2 | 98.1 ±0.6 | 98.8 ±0.7 | 99.0 ±0.5 |
| **L0 + BERT + LF** | no | 97.8 ±0.7 | 97.5 ±0.7 | 98.7 ±0.5 | 98.4 ±0.5 | 98.4 ±0.6 |
| **BERT + LFx2** | no | 98.0 ±1.2 | 97.9 ±0.9 | 99.0 ±0.8 | 98.8 ±1.0 | 98.3 ±0.8 |
| **BERT** | yes | 97.5 ±0.6 | 98.3 ±0.7 | 98.5 ±0.6 | 98.4 ±0.2 | 98.7 ±0.6 |
| **L0 + BERT** | yes | 97.8 ±0.4 | 98.1 ±0.2 | 98.7 ±0.5 | 98.7 ±0.2 | 98.7 ±0.5 |
| **BERT + LF** | yes | 98.3 ±0.5 | 98.6 ±0.6 | 98.7 ±0.7 | 98.6 ±0.4 | 99.1 ±0.4 |
| **L0 + BERT + LF** | yes | 98.7 ±0.5 | 98.1 ±0.7 | 98.9 ±0.8 | 98.4 ±0.1 | 98.8 ±0.7 |
| **BERT + LFx2** | yes | 98.2 ±0.6 | 97.5 ±0.3 | 98.1 ±0.5 | 97.9 ±0.5 | 98.5 ±0.4 |

Table 8: **German Intent Classification**: Results for German BERT with 95% confidence interval measured for five trials, evaluated on the standard German test data. All models maintain performance on the standard text, as desired.