# OpenReview forum: "BERTwich: Extending BERT’s Capabilities to Model Dialectal and Noisy Text"
_EMNLP/2023/Conference — EMNLP 2023 Findings_

### Official Review · Reviewer_FYgr · 2023-08-02

**Soundness:** 2

**Excitement:**

2: Mediocre: This paper makes marginal contributions (vs non-contemporaneous work), so I would rather not see it in the conference.

**Missing References:**

Prior works for introducing noise during pretraining and finetuning:

Mike Lewis, Yinhan Liu, Naman Goyal, Marjan Ghazvininejad, Abdelrahman Mohamed, Omer Levy, Veselin Stoyanov, and Luke Zettlemoyer. 2020. BART: Denoising Sequence-to-Sequence Pre-training for Natural Language Generation, Translation, and Comprehension. In Proceedings of the 58th Annual Meeting of the Association for Computational Linguistics, pages 7871–7880, Online. Association for Computational Linguistics.

Ziang Xie, Guillaume Genthial, Stanley Xie, Andrew Ng, and Dan Jurafsky. 2018. Noising and Denoising Natural Language: Diverse Backtranslation for Grammar Correction. In Proceedings of the 2018 Conference of the North American Chapter of the Association for Computational Linguistics: Human Language Technologies, Volume 1 (Long Papers), pages 619–628, New Orleans, Louisiana. Association for Computational Linguistics.

Wu, Z., Papadimitriou, I., & Tamkin, A. (2022). Oolong: Investigating What Makes Crosslingual Transfer Hard with Controlled Studies. ArXiv, abs/2202.12312.

Paul Michel and Graham Neubig. 2018. MTNT: A Testbed for Machine Translation of Noisy Text. In Proceedings of the 2018 Conference on Empirical Methods in Natural Language Processing, pages 543–553, Brussels, Belgium. Association for Computational Linguistics.

Vaibhav Vaibhav, Sumeet Singh, Craig Stewart, and Graham Neubig. 2019. Improving Robustness of Machine Translation with Synthetic Noise. In Proceedings of the 2019 Conference of the North American Chapter of the Association for Computational Linguistics: Human Language Technologies, Volume 1 (Long and Short Papers), pages 1916–1920, Minneapolis, Minnesota. Association for Computational Linguistics.

Prior works for modular LLM architectures to handle linguistic variation:

Jonas Pfeiffer, Ivan Vulić, Iryna Gurevych, and Sebastian Ruder. 2020. MAD-X: An Adapter-Based Framework for Multi-Task Cross-Lingual Transfer. In Proceedings of the 2020 Conference on Empirical Methods in Natural Language Processing (EMNLP), pages 7654–7673, Online. Association for Computational Linguistics.

William Held, Caleb Ziems, and Diyi Yang. 2023. TADA : Task Agnostic Dialect Adapters for English. In Findings of the Association for Computational Linguistics: ACL 2023, pages 813–824, Toronto, Canada. Association for Computational Linguistics.

**Paper Topic And Main Contributions:**

This paper aims to make a Pretrained Language Model more robust to naturally occurring noise and linguistic variation. To do so, they propose adding additional layers to the start and end of a pretrained language model, using continued pretraining on data which has been synthetically noised using character level noise patterns, and finetuning on similarly noised data. They find that this full procedure helps with English typos and with a Bavarian dialect of German, while only finetuning on noisy data benefits performance in Swiss German. Finally, they perform an analysis of token-level alignment throughout the original and modified models and how this correlates with classifiers trained at different layers of the model as well.

**Questions For The Authors:**

A) Are there patterns to the cases in German dialects which switch from wrong to right with noisy pretraining? What types of linguistic patterns does this noisy training capture? Maybe this could help explore why the Swiss German performance degrades when noise is introduced.

B) How do you explain the higher similarity at the start of BERT+LF than of L0+BERT?

C) What were the hyperparameters used for experiments? Learning rate, Batch Size, epochs, etc

**Reasons To Accept:**

- Assessing the benefits of introducing synthetic noise during both pretraining and finetuning is an interesting question. Understanding how much general robustness can be derived from pretraining on synthetically noised data alone might yield good insights into the ways in which models maintain more broad linguistic knowledge.

- Assessing the effects of the procedure on multiple dialects of German is valuable, since it allows experiments which tease apart the benefits depending on the types of variation which occurs in each dialect.

**Reasons To Reject:**

- The primary novelty of the work appears to be the architectural change, but they don't compare to other widely used architecture modifications for transformers such as Adapters, Prefix-Tuning, or LoRA. The baselines (L0, LF, and regular BERT) all have strictly fewer tunable parameters than the proposed approach, so it seems totally possible to me that the improved results of the sandwich approach are simply from the increased number of parameters especially since the sandwich approach frequently does not outperform even these baseline architectures by a statistically significant margin. It seems unclear that this architecture is a meaningful improvement without more diverse baselines, especially ones which control for parameter count.

- The approach for synthetic introduction of noise (insertion, deletion, replacement, and swapping) primarily address typos with some coverage of low-edit distance dialectal morphology changes. This is a quite small subset of dialectal shifts compared to full lexical shift and syntactic variation. Furthermore, even for typo noise this model is somewhat naive and assigns even probability to noise categories, despite real noise following patterns e.g. the example "student" -> "studebt" seems reasonable because B and N are neighbors on the keyboard, but "student" -> "studeqt" would almost never happen. This work would be much stronger if it incorporated and compared to synthetic noise grounded in actual typo patterns and language variation.

- The model omits many standard details of training which would make it difficult to reproduce.

**Reproducibility:**

3: Could reproduce the results with some difficulty. The settings of parameters are underspecified or subjectively determined; the training/evaluation data are not widely available.

**Reviewer Confidence:**

5: Positive that my evaluation is correct. I read the paper very carefully and I am very familiar with related work.

**Typos Grammar Style And Presentation Improvements:**

Figure 1: This figure is quite large, but with not a ton of information. This could be shrunk significantly (stack the 12 internal layers!) to make room for key information such as the settings for both CPT and finetuning training runs.

Line 226: Repeated List of Operations twice

Line 394-396: "models maintain performance on Standard" - Where do you show this? It doesn't appear to be in any of the tables in the paper?

---

> ### Author Rebuttal · Authors · 2023-08-29
>
> Thank you very much for your thorough review.  We appreciate your thoughtful suggestions and the opportunity to answer your questions.  We also appreciate the missing references you listed, and we will incorporate them into our related work section – and elsewhere when appropriate – in the final version of the paper.
>
> A) Are there patterns to the cases in German dialects which switch from wrong to right with noisy pretraining? What types of linguistic patterns does this noisy training capture? Maybe this could help explore why the Swiss German performance degrades when noise is introduced.
>
> Upon your suggestion, we looked at the examples the model got incorrect for the baseline (BERT without noise) and the best BERTwich model for both South Tyrolean and Swiss German.  The takeaway is one we would have suspected – many of the examples switch from wrong to right (as evidenced by the jump in accuracy); however, in the examples that do not switch, several of the lexical items are very different from those that would appear in the standard version of the sentence.  The mutual comprehensibility for these examples would likely be far lower and would require extra knowledge of new vocabulary items.  We would like to conduct this sort of error analysis in more detail and perhaps include some insights in the final version of the paper.
>
> We would also note that the Swiss German performance improves when noise is introduced in fine-tuning.  Performance is also much better in the BERTwich models than simply fine-tuning German BERT on standard German.  But it is true that German BERT fine-tuned with 40% noise (with no architectural modifications) yields the highest performance over BERTwich models.
>
> B) How do you explain the higher similarity at the start of BERT+LF than of L0+BERT?
>
> The results you reference (Table 5) demonstrate that if it’s between L0 or LF, LF is more powerful; that is, L0 is not as helpful for adaptation to nonstandard text on its own.  Rather, the utility of L0 is unlocked when working in conjunction with LF, and this combination yields the best performance and much higher cosine similarity scores in the embedding space between un-noised and noised words. We suspect that this relates to past work like “BERT Rediscovers the Classical NLP Pipeline” (Tenney et al., 2019), in that L0 is useful for learning lower-level patterns like orthographic changes, but interpretable changes to the embedding space and other measures of language modeling capability do not make sense without including LF for CPT.  This way, higher-level patterns can be captured in the top layer with greater influence on the language modeling head in the architecture.
>
> C) What were the hyperparameters used for experiments? Learning rate, Batch Size, epochs, etc.
>
> We will certainly include this information in our experiments section (Section 4) in the final version.  In all of our training runs, we use the AdamW optimizer and a minibatch size of 8.  When doing the continued pre-training of a model, we set the learning rate to 1e-4.  When fine-tuning a model, we set the learning rate to 1e-5.  As described in our Experiments section, we vary the random seed across trials and report the average across trials.
>
> To respond to your comment, “they don't compare to other widely used architecture modifications for transformers such as Adapters, Prefix-Tuning, or LoRA”:
>
> We took your suggestion and ran an experiment in which we use the LoRA approach to adapt BERT to the SST task, and the fine-tuning data we use here is the noisy SST data, as was done in the paper with vanilla fine-tuning.  Our results demonstrate that using the LoRA approach with noisy data performs similarly to vanilla fine-tuning of BERT with noisy data across noise levels.  The highest performance attained using LoRA with BERT and our noisy FT approach was 68%, which is still lower than the highest performance attained by BERTwich (74.3%).  We will perform more thorough experiments with LoRA and include the results in the final version of the paper as another baseline.
>
> We considered using adapters earlier in the project, but we thought the first step would be to see what types of improvements we can get by making simpler, more modular changes to the architecture in the hopes of having higher interpretability (for example, the type of analyses we do in Table 4 and 5).
>
> To respond to your comment, “This work would be much stronger if it incorporated and compared to synthetic noise grounded in actual typo patterns and language variation (e.g., full lexical shift and syntactic variation)”:
>
> We would like to mention something found in our preliminary experiments that led us to make the decision to include random character-level noise rather than character-level noise based on frequent typos or instances of phonological variation.  We initially considered specific character swaps as our synthetic noise in training, where these swaps were based on the QWERTY keyboard (i.e., only swap X with a letter that is adjacent to X on the keyboard) or common phonological replacements (e.g., ‘r’ and ‘l’).  Anecdotally speaking, we did not find this to change the results, and this finding is echoed in previous work (Aepli & Sennrich, 2022).  We believe this is because our goal in applying character-level noise was to disrupt the subword tokenization, such that over-segmentation and subword replacement are artificially induced, exposing the model to varying tokenized versions of the same words or phrases.  In this respect, the introduction of character-level noise serves more as a means to expose the model to these phenomena during training rather than to mirror expected or potential test examples.
>
> To respond to your comment, “‘models maintain performance on Standard’ - Where do you show this? It doesn't appear to be in any of the tables in the paper?”:
>
> As part of our experimental runs, we also recorded the performance on the standard versions of the task; that is, performance of the English models on the original SST data, and performance of the German models on standard German itself.  However, the scores were all very close (no statistically significant differences) regardless of the model variant used, so we did not include these results in the paper so as to save space.  However, your point is certainly a good one that it would be important to explicitly highlight these results for completeness; we will include them in the appendix of the final version.
>
> To respond to your comments, “it seems totally possible to me that the improved results of the sandwich approach are simply from the increased number of parameters” and “unclear that this architecture is a meaningful improvement without more diverse baselines, especially ones which control for parameter count”:
>
> Though we do not have enough time to perform the experiment before the end of the author response period, we could try applying the BERTwich method to the 10-layer BERT model released by Google; that way, the number of parameters in this BERTwich model would be equivalent to that of the original 12-layer BERT-base.  However, this would unfairly favor the baseline, because the BERT models from Google have undergone full pre-training, while our extra layers learn only from a fraction of the data during CPT. As such, they serve the purpose of extending the modeling capabilities (much like fine-tuning in this respect), but they are not powerful enough to take over the job of the original encoder stack – that is, without sufficient pre-training, the increased number of parameters alone is not enough to yield the large increases in performance demonstrated by our results.  The experiment with 10-layer BERT can be added to provide another way of interpreting the results.
>
> Once again, many thanks for your review!

---

### Official Review · Reviewer_FFsj · 2023-08-04

**Soundness:** 3

**Excitement:**

3: Ambivalent: It has merits (e.g., it reports state-of-the-art results, the idea is nice), but there are key weaknesses (e.g., it describes incremental work), and it can significantly benefit from another round of revision. However, I won't object to accepting it if my co-reviewers champion it.

**Paper Topic And Main Contributions:**

This paper tries to improve BERT's language modelling when dealing with noisy data. The noisy input could be either due to misspelling/grammatical error, or when a dialect of language is parsed instead of the high-resource language that the model is trained on.
The authors propose adding two layers to the BERT's architecture and continue pre-training on character-level noisy data from Wikipedia to help the LM better handle noisy inputs. They show that the proposed method will enhance generalization (compared to baselines like BERT without adding extra layers) to noisy/non-standard data for two tasks: 1) noisy SST test set for English BERT 2) intent classification on Swiss-german and Tyrolean (German dialects) for German BERT.
The generalisation especially becomes better when the fine-tuning is performed with higher noise in fine-tuning data (for both the method and baselines). They also show that in the proposed model cosine similarity between a given word and its noisy variant becomes higher compared to baseline models.

**Questions For The Authors:**

- A suggestion: I propose experimenting with PEFT methods like adapters, LoRA, etc. as well, which addresses the "fairness" in comparison, as the introduced parameters to the models are relatively small.

**Reasons To Accept:**

- The idea of improving generalisation to close dialects by introducing synthetic noise to pre-training/fine-tuning phase is an interesting approach for dealing with low-resource languages (even though it has been previously studied on e.g. PoS tagging and several sequence classification tasks.
- The analysis on the similarity between words and their noisy variants is quite interesting, and gives an intuition why pre-training/fine-tuning considerably improves generalisation to noisy/non-standard data.


**Reasons To Reject:**

- Although the improvements over baselines seems interesting, comparing BERT with a model that has two more layers, due to considerably increasing the number of parameters doesn't seem to be a fair comparison. After all, the improvement over BERT when CPT+40% is applied is not that significant (even in Table 3 the "naive" BERT seems to be the best performing model).
- All the experiments are done with BERT which haven't seen any "informal" data during pre-training. A model like RoBERTa, which is also pre-trained on e.g. OpenWebText (with possibly noisy data), would have been a natural candidate for all the experiments instead of adding extra layers to an under-trained model.
- The main novelty of the paper seems to be proposing adding two layers to the architecture, as the noisy pre-training/fine-tuning seems to be proposed by previous work.


**Reproducibility:**

3: Could reproduce the results with some difficulty. The settings of parameters are underspecified or subjectively determined; the training/evaluation data are not widely available.

**Reviewer Confidence:**

3: Pretty sure, but there's a chance I missed something. Although I have a good feel for this area in general, I did not carefully check the paper's details, e.g., the math, experimental design, or novelty.

**Typos Grammar Style And Presentation Improvements:**

- line 226: repetitive text

---

> ### Author Rebuttal · Authors · 2023-08-29
>
> Thank you very much for your review and thoughtful suggestions.
>
> To address your question, "A suggestion: I propose experimenting with PEFT methods like adapters, LoRA, etc. as well, which addresses the "fairness" in comparison, as the introduced parameters to the models are relatively small.":
>
> We took your suggestion and ran an experiment in which we use the LoRA approach to adapt BERT to the SST task, and the fine-tuning data we use here is the noisy SST data, as was done in the paper with vanilla fine-tuning.  Our results demonstrate that using the LoRA approach with noisy data performs similarly to vanilla fine-tuning of BERT with noisy data across noise levels.  The highest performance attained using LoRA with BERT and our noisy FT approach was 68%, which is still lower than the highest performance attained by BERTwich (74.3%).  We will perform more thorough experiments with LoRA and include the results in the final version of the paper as another baseline.
>
> We considered using adapters earlier in the project, but we thought the first step would be to see what types of improvements we can get by making simpler, more modular changes to the architecture in the hopes of having higher interpretability (for example, the type of analyses we do in Table 4 and 5).
>
> To respond to your comment, “All the experiments are done with BERT which haven't seen any "informal" data during pre-training. A model like RoBERTa, which is also pre-trained on e.g. OpenWebText (with possibly noisy data), would have been a natural candidate for all the experiments instead of adding extra layers to an under-trained model.”:
>
> We did preliminary experiments on BERT, RoBERTa, and CharBERT, fine-tuning on a sentiment analysis task and testing on synthetically noised test examples (a baseline experiment in our paper).  We found that all three models had similarly low performance across various synthetic noise settings.  While CharBERT is a character-based model and may provide some advantage when dealing with traditionally over-segmented text, and as you rightly said, RoBERTa includes some informal text in its training data and uses BPE tokenization (BPE is often better than WordPiece when dealing with text that does not match the training data), we selected the most “vanilla” of the models when it comes to being specialized for the issue we are investigating - BERT.  This way, we could specifically see the effects of our approach to optimizing the model from scratch, and we are doing zero-shot adaptation.  We would like to add results for RoBERTa and/or BART for the final version of the paper for comparison.
>
> To respond to your comment, “Although the improvements over baselines seems interesting, comparing BERT with a model that has two more layers, due to considerably increasing the number of parameters doesn't seem to be a fair comparison. After all, the improvement over BERT when CPT+40% is applied is not that significant (even in Table 3 the "naive" BERT seems to be the best performing model).”:
>
> Though we do not have enough time to perform the experiment before the end of the author response period, we could try applying the BERTwich method to the 10-layer BERT model released by Google; that way, the number of parameters in this BERTwich model would be equivalent to that of the original 12-layer BERT-base.  However, this would unfairly favor the baseline, because the BERT models from Google have undergone full pre-training, while our extra layers learn only from a fraction of the data during CPT. As such, they serve the purpose of extending the modeling capabilities (much like fine-tuning in this respect), but they are not powerful enough to take over the job of the original encoder stack – that is, without sufficient pre-training, the increased number of parameters alone is not enough to yield the large increases in performance demonstrated by our results.  The experiment with 10-layer BERT can be added to provide another way of interpreting the results.
>
> Finally, we would like to mention details of our experimental setup for reproducibility (i.e., hyperparameters):
> In all of our training runs, we use the AdamW optimizer and a minibatch size of 8.  When doing the continued pre-training of a model, we set the learning rate to 1e-4.  When fine-tuning a model, we set the learning rate to 1e-5.  As described in our Experiments section, we vary the random seed across trials and report the average across trials.  We will certainly include this information in our experiments section (Section 4) in the final version of our paper.
>
> Once again, many thanks for your review!

---

### Official Review · Reviewer_x8AR · 2023-08-12

**Soundness:** 3

**Excitement:**

4: Strong: This paper deepens the understanding of some phenomenon or lowers the barriers to an existing research direction.

**Paper Topic And Main Contributions:**

The paper mainly focuses on exploring performance issues of fine-tuned BERT on nonstandard text (eg: dialects, misspellings etc). The paper introduces the idea of adding randomly initialized layers to the encoder stack of pre-trained BERT (BERTwich) trained on noisy data to improve BERT’s performance on nonstandard text. BERTwich was evaluated mainly on 3 tasks across different variations of the proposed approach. The authors also found that BERTwich reduced the distance between the words and their noisy counterparts in the embedding space thereby expanding the modeling capabilities of BERT.

**Questions For The Authors:**

a. Is there a reason why noise levels above 40% were not evaluated?

b. Authors should consider including the performance of BERTwich on standard tasks along with the respective baselines. Additionally, quantifying the additional compute overhead of BERTwich will be helpful.

c. Results on other variants of BERT and more datasets will help in strengthening the paper further.

**Reasons To Accept:**

- The paper is well written and easy to follow
- The authors perform many ablation experiments across different model variations, with/without continued pre-training and different noise levels to understand the impact of each component
- BERTwich seems to out-perform the baseline BERT model in most cases when fine-tuned with noisy data.
- The paper provides interesting insights to understand how the proposed modifications helps in increasing the cosine similarities between words and their noisy counterparts in the embedding space

**Reasons To Reject:**

- While the authors claim that performance of BERTwich on standard tasks were not impacted with the proposed modifications, its important to highlight these results in the paper. Currently results on standard versions of different tasks along with the baseline are missing
- Authors have not quantified the additional computational overhead with continued pre-training and architectural modifications
- Evaluation on other variations of BERT (BERT-Large, DistilBERT etc)  and other datasets (Italian/Spanish intent classification) are missing
- Comparisons to past work are not clearly mentioned in the results




**Reproducibility:**

3: Could reproduce the results with some difficulty. The settings of parameters are underspecified or subjectively determined; the training/evaluation data are not widely available.

**Reviewer Confidence:**

4: Quite sure. I tried to check the important points carefully. It's unlikely, though conceivable, that I missed something that should affect my ratings.

---

> ### Author Rebuttal · Authors · 2023-08-29
>
> Thank you very much for your insightful comments and for the opportunity to answer your questions.
>
> a) Is there a reason why noise levels above 40% were not evaluated?
>
> We didn't try noise levels of 50% or more because it seemed too extreme to disrupt a majority of letters. We could certainly add these experiments in the final version, although even if they perform better, we would be reluctant to recommend such an extreme amount of noise for general use.
>
> b) Authors should consider including the performance of BERTwich on standard tasks along with the respective baselines. Additionally, quantifying the additional compute overhead of BERTwich will be helpful.
>
> As part of our experimental runs, we also recorded the performance on the standard versions of the task; that is, performance of the English models on the original SST data, and performance of the German models on standard German itself.  However, the scores were all very close (no statistically significant differences) regardless of the model variant used, so we did not include these results in the paper so as to save space.  However, your point is certainly a good one that it would be important to explicitly highlight these results for completeness; we will include them in the appendix of the final version.
>
> The additional compute overhead of BERTwich (specifically, doing CPT for L0+BERT+LF) is about 3.4 days on a single NVIDIA Tesla P100 GPU.  We will include this information in the final version of the paper.
>
> c) Results on other variants of BERT and more datasets will help in strengthening the paper further.
>
> We did preliminary experiments on BERT, RoBERTa, and CharBERT, fine-tuning on a sentiment analysis task and testing on synthetically noised test examples (a baseline experiment in our paper).  We found that all three models had similarly low performance across various synthetic noise settings.  We selected the most “vanilla” of the models when it comes to being specialized for the issue we are investigating - BERT.  This way, we could specifically see the effects of our approach to optimizing the model from scratch, so to speak.  We would like to add results for other variants of the model, such as DistilBERT, RoBERTa, and/or BART, for the final version of the paper for comparison.
>
> Unfortunately, it is quite difficult to come by data that includes multiple dialects and is labeled for a specific NLP task like sequence classification, rather than masked language modeling or dialect identification.  We spent a lot of time looking for appropriate data and found the German intent classification dataset to be a good candidate.  We also have access to Italian intent classification (with one dialect of Italian, Neapolitan); we can certainly add the results for Neapolitan in the final version of our paper.  In addition, just as we applied synthetic noise to the English SST dataset for testing, we could do the same for a few more tasks (e.g., from the GLUE benchmark) for the final version of the paper.
>
> To respond to your comment, “Comparisons to past work are not clearly mentioned in the results”:
>
> We directly compare to two pieces of past work in our results: 1) BERT with CPT and without fine-tuning noise comes from Aepli and Sennrich (2022), and 2) BERT without CPT and with increasing fine-tuning noise comes from Srivastava and Chiang (2023).  We attempt to explain this in Section 4.2 of our paper.  We can make it clearer in the results tables itself for the final version.
>
> Finally, we would like to mention details of our experimental setup for reproducibility (i.e., hyperparameters):
> In all of our training runs, we use the AdamW optimizer and a minibatch size of 8.  When doing the continued pre-training of a model, we set the learning rate to 1e-4.  When fine-tuning a model, we set the learning rate to 1e-5.  As described in our Experiments section, we vary the random seed across trials and report the average across trials.  We will certainly include this information in our experiments section (Section 4) in the final version of our paper.
>
> Once again, many thanks for your review!

---

### Meta-Review · Area_Chair_CSFv · 2023-09-23

**Recommendation:** 3

**Metareview:**

This paper explores using a sandwich architecture around BERT (a BERTwich) to improve performance for non-standard text. Experiments are performed in English and Swiss German to validate performance, and ablation studies are performed to determine the importance of the two halves of the sandwich.

The discussion period was very active, and while there are some comparison experiments reported in the discussion that perhaps should have been in the original paper, the reviewers generally found the work sound (scores of 3/3/4, note that reviewer FYgr said they would increase their score to 4 but could not successfully edit it, so I reflect that here). Excitement for the approach varied more (2/3/4), especially given the strong performance of prior work on this task when evaluated appropriately (see next paragraph).

The authors reported in the discussion period that there may not be an advantage to this approach over the better-known LoRA. However, I do believe there is still a benefit to publishing this work as it provides an alternative approach that in future experiments or different settings may provide an advantage.

When revising the paper, the authors must add the experiments described in the discussion, and it would be very beneficial to try to find any benefit to this approach over LoRA, either quantitatively or qualitatively. The authors may want to consider whether the focus of the paper should be on methods for success in the nonstandard text tasks they explore (exploring both LoRA and sandwich approaches with equal interest) or whether they still want to focus primarily on the sandwich approach.

---

### Decision · Program_Chairs · 2023-10-07

**Decision:**

Accept-Findings

**Comment:**

This paper explores using a sandwich architecture around BERT (a BERTwich) to improve performance for non-standard text. Experiments are performed in English and Swiss German to validate performance, and ablation studies are performed to determine the importance of the two halves of the sandwich.

The discussion period was very active, and while there are some comparison experiments reported in the discussion that perhaps should have been in the original paper, the reviewers generally found the work sound (scores of 3/3/4, note that reviewer FYgr said they would increase their score to 4 but could not successfully edit it, so I reflect that here). Excitement for the approach varied more (2/3/4), especially given the strong performance of prior work on this task when evaluated appropriately (see next paragraph).

The authors reported in the discussion period that there may not be an advantage to this approach over the better-known LoRA. However, I do believe there is still a benefit to publishing this work as it provides an alternative approach that in future experiments or different settings may provide an advantage.

When revising the paper, the authors must add the experiments described in the discussion, and it would be very beneficial to try to find any benefit to this approach over LoRA, either quantitatively or qualitatively. The authors may want to consider whether the focus of the paper should be on methods for success in the nonstandard text tasks they explore (exploring both LoRA and sandwich approaches with equal interest) or whether they still want to focus primarily on the sandwich approach.